# Social and professional recognition are key determinants of quality of life at work among night-shift healthcare workers in Paris public hospitals (AP-HP ALADDIN COVID-19 survey)

Martin Duracinsky[1,2,3], Fabienne Marcellin[4]*, Lorraine Cousin[2,3,4], Vincent Di Beo[4], Véronique Mahé[5], Olivia Rousset-Torrente[2,3], Patrizia Carrieri[4], Olivier Chassany[2,3]

1 Département de Médecine Interne et d'immunologie Clinique, Hôpital Bicêtre, AP-HP, Kremlin-Bicêtre, France, 2 Patient-Reported Outcomes Unit (PROQOL), UMR 1123, Inserm, Université de Paris, Paris, France, 3 Unité de Recherche Clinique en Économie de la Santé (URC-ECO) AP-HP, Hôpital Hôtel-Dieu, Paris, France, 4 Inserm, IRD, SESSTIM, Sciences Economiques & Sociales de la Santé et Traitement de l'Information Médicale, ISSPAM, Aix Marseille Univ, Marseille, France, 5 Service Central de Santé au Travail, Hôpitaux Lariboisière-Fernand Widal, AP-HP Nord, Paris, France

* fabienne.marcellin@inserm.fr

**Data Availability Statement:** Given the confidential and sensitive nature of data collected (including

## Abstract

### Objective

Documenting the perceptions and experiences of frontline healthcare workers during a sanitary crisis is key to reinforce healthcare systems. We identify the determinants of quality of working life (QWL) among night-shift healthcare workers (NSHW) in Paris public hospitals shortly after the first-wave of the COVID-19 pandemic.

### Methods

The ALADDIN cross-sectional online survey (15 June to 15 September 2020) collected QWL, socio-economic, behavioral, and work-related information among 1,387 NSHW in the 39 hospitals of the Assistance Publique—Hôpitaux de Paris (AP-HP). Data were weighted (margin calibration) to be representative of the entire population of 12,000 AP-HP hospitals' NSHW regarding sex, age, and professional category. Linear regression was used to identify correlates of QWL (WRQoL scale).

### Results

New night position during the COVID pandemic, difficulties in getting screened for COVID, and considering protective measures inadequate were associated with poorer QWL, after adjustment for socio-economic characteristics, professional category, perceived health, physical activity, and history of harassment at work. Under-estimation of night-shift work by day-shift colleagues, reporting night work as a source of tension with friends, or feeling more irritable since working at night also impaired QWL. By contrast, satisfaction regarding COVID information received from the employer, and feeling valued by the general population during the pandemic improved QWL.

data on use of illicit psychoactive substances) and the presence of potentially identifying information, the minimal data set cannot be shared in a public repository. Sharing restrictions are imposed by the Scientific committee of the survey. Data are available upon request to the scientific committee of the ALADDIN survey (contact: proqol. research@gmail.com). Requests sent to proqol. research@gmail.com will be processed by the Scientific committee of the survey.

**Funding:** This work was supported by the Assistance Publique-Hôpitaux de Paris (AP-HP) Foundation, the French Institute of Research in Public Health (IReSP), and the National Institute of Cancer (INCa) 2018 grant for doctoral students.

**Competing interests:** The authors have declared that no competing interests exist.

## Conclusions

Insufficient access to screening, information, and protective measures impaired QWL of NSHW after the first wave of COVID-19 in Paris public hospitals. Social and professional recognition of night-shift work were the key determinants of QWL in this population.

## Introduction

The number of individuals involved in night-shift work has increased in the last twenty years in Western countries, and people working in the healthcare sector are among the most represented [1–3]. Night-shift work negatively impacts health, notably because of circadian rhythms perturbations [4–8]. Its multiple potential consequences include fatigue, sleep disturbance, an increased risk of cancer, cardiovascular diseases, metabolic syndrome, affective disorders, and impaired cognitive function [9–19]. Beyond these negative health consequences, night-shift work has also detrimental effects on quality of life [20–22], which closely interacts with perceived health. Quality of working life (QWL), a multidimensional definition of well-being in the workplace, directly influences quality of life [23]. QWL is closely linked to factors such as workload, work-life balance, meaning of work and meaning at work [24, 25], the latter two factors distinguishing how one individual perceives the meaning of what one does at work from the group to which one self-identifies in said work environment [26]. A recent literature review targeting healthcare work showed that QWL may also influence quality of care [27]. In the COVID-19 era, QWL remains poorly documented among hospital night-shift healthcare workers (NSHW), a population exposed to a higher risk of infection [28], and who contributes to the continuity of care since the beginning of the pandemic. Managing healthcare systems during sanitary crises represents human and organizational challenges with potential mental health and quality of life implications [29]. A better understanding of QWL among NSHW is therefore needed, both from a public health perspective, and to identify levers which could help strengthening healthcare systems in such contexts.

The present study aims to document QWL among NSHW in Paris public hospitals shortly after the first-wave of the COVID-19 pandemic and to identify its determinants.

## Materials and methods

### The AP-HP ALADDIN survey

The ALADDIN cross-sectional survey (15 June 2020 to 15 September 2020) was conducted among NSHW in public hospitals in Paris. It included all 39 hospitals of the Assistance Publique—Hôpitaux de Paris (AP-HP). One of the main objectives was to document NSHW's QWL (i.e. perceived quality of life at work) and its correlates shortly after the first wave of the COVID-19 pandemic (March to May 2020), once healthcare workers were more available to participate in the survey. All individuals working in the AP-HP hospitals with a night-shift or a day/night alternation employment contract, regardless the years of experience, working full-time or part-time, could participate in the survey. Exclusion criteria were as follows: (i) working only during the day; (ii) working less than three hours a day between 9 p.m. and 6 a.m. twice a week (including on-duty or on-call staff). In order to maintain a homogeneous study population, physicians were excluded from the analyses as they constitute a subgroup with specific characteristics. Sample size was expected to reach 10% of the 12,000 NSHW working in the AP-HP hospitals (target population).

## Ethics

The AP-HP ALADDIN survey was approved by the Lyon 2 ethics committee in March 2020 (ID RCB202-A00495-34). Informed consent was obtained for all survey participants.

## Data collection

During the AP-HP ALADDIN survey, quantitative data were collected using an online questionnaire which documented participants' sociodemographic, economic and work-related characteristics, perceived health, QWL, as well as perceptions and experience since the beginning of the COVID-19 pandemic [30, 31]. NSHW's perceptions regarding their social and professional recognition were assessed using items related to under-estimation of night-shift work by colleagues, loved ones, and patients; perceptions of the importance of night missions and of workload during night; feeling valued by the general population as a NSHW during the pandemic. Most of these items were derived from different stigma scales [32–34]. NSHW could respond to the questionnaire online (NetSurvey®), using either their computer at work or their personal electronic devices.

## Assessment of QWL

NSHW's QWL was assessed using the work-related quality of life (WRQoL) scale [35] which includes 24 items, each associated with five possible answers on a Likert-type scale (strongly disagree/disagree/neutral/agree/strongly agree). The WRQoL scale explores six dimensions of quality of life related to the work environment of NSHW. Each dimension is associated with a factor score, calculated from respondents' answers to the first 23 items of the scale: general well-being (GWB, score range: 0 to 30); home-work interface (HWI, score range: 0 to 15); job and career satisfaction (JCS, score range: 0 to 30); control at work (CAW, score range: 0 to 15); working conditions (WCS, score range: 0 to 15); and stress at work (SAW, score range: 0 to 10) [36]. For each dimension, higher score values denote better QWL. A full-scale score, ranging from 0 to 115, can also be calculated as the sum of the six factor scores. The 24th item of the scale, which explores NSHW's satisfaction with the overall quality of their working life, is not used in the calculation of factor scores.

## Study population

The study population included survey participants who filled out the WRQoL scale.

## Statistical analyses

Data were weighted and calibrated (calibration on margins using the raking ratio method) to be representative of the whole population of the 12,000 NSHW working in the AP-HP hospitals in terms of sex, age (using 5-year age classes), and professional category (nurses, nurse assistants and laboratory technicians, executives, midwives, and other categories). Descriptive statistics were used to document NSHW's answers to the questionnaire items and the distribution of QWL scores in the whole study population. Comparisons were then performed between professional categories using chi-square tests for categorical variables and Wald tests for continuous ones. Lastly, weighted linear regression models were used to identify correlates of the WRQoL full-scale score. Variables with a p-value $<0.25$ in the univariable analyses were considered eligible for the multivariable model. A backward selection procedure was used to build the final multivariable model, which included only statistically significant variables ($p<0.05$). The Stata version 14.2 for Windows software (StataCorp, College Station, Texas, USA) was used for the analyses.

## Results

### Characteristics of the study population

The study population included 1,387 individuals, and mainly comprised nurses (52.3%). The other professional categories represented were nurse assistant or technicians (38.2%), midwives (4.2%), executives (0.8%), and other categories (4.6%). The latter group included different professions such as reception agents, administrative staff, or pharmacists.

NSHW in the study population were mostly women (77.5%). Mean age (standard deviation, SD) was 39.3 (12.0 years), 54.2% of NSHW were living with a partner, and 50.2% had children (Table 1A). Fourteen percent of NSHW reported facing financial difficulties. Three quarters (75.8%) had a permanent night position and 61.2% worked in a hospital department for adult care. The mean (SD) seniority as a night-shift worker was 9.0 (8.5) years (Table 1B). Sociodemographic, economic, and work-related characteristics differed significantly between professional categories of NSHW (Table 1A to 1E).

**Health-related characteristics.** In the whole study population, 51.3% perceived their health as good or excellent, and 54.2% reported physical activity, with highest rates among midwives for these two characteristics (Table 1C). Twenty-one percent (20.9%) of NSHW had faced sexual or moral harassment at work. Nearly fourteen percent of NSHW (13.6%) reported they had contracted COVID-19, but 27.7% did not answer the corresponding item of the questionnaire.

**Work-related perceptions.** Globally, 64.7% of NSHW perceived that night-shift work was often or always under-estimated by their colleagues working during day, and this percentage was highest among executives (Table 1D). Concerning the social consequences of work, 47.2% of NSHW with a partner or children reported work rhythm was a source of tension between one another, 20.1% of NSHW reported work rhythm as a source of tension with friends, and 43.6% felt more irritable since they worked at night. The percentages were highest among midwives for these three characteristics.

**Changes in work organization since the beginning of the COVID-19 pandemic.** Regarding work organization, 36.8% of NSHW reported no change since the beginning of the COVID-19 pandemic, a percentage that was highest in the "other categories" group (Table 1E). Both globally and almost consistently within professional categories, the changes more often reported were increase in the number of working hours (globally 37.2% of NSHW) and change of ward (part of department dedicated to a given specialty) (29.8%). Less than one percent (0.8%) had a new night-shift position since the beginning of COVID-19. Nineteen percent of NSHW changed activity to manage COVID-19 patients, and this percentage was highest among nurses.

**COVID-related items.** NSHW's responses to the COVID-19 items of the questionnaire showed that most NSHW (77.8%) felt vulnerable to COVID-19 because of their professional activity and 90.6% of them feared to transmit the virus to close relatives (Table 1F). About one third (31.5%) reported that the information their employer gave them on COVID-19 was sufficient and complete, 58.4% faced difficulties in getting screened, 59.7% reported difficulties in applying protective measures against COVID-19, which 27.6% considered inadequate. A total of 19.6% of NSHW felt confident in the health authorities' ability to manage the crisis. Finally, while 62.9% felt valued by the general population as a NSHW during the pandemic, 7.0% and 8.4% reported having received recent psychological support from close relatives and professionals, respectively (Table 1). No significant difference between professional categories were found concerning NSHW's responses to the COVID-related items of the questionnaire, except for the percentage of NSHW who felt vulnerable to COVID-19 due to their professional activity (highest among midwives) and that facing difficulties in getting screened for SARS-CoV-2 infection (highest among assistant nurses or technicians).

**Table 1.** Main characteristics of night-shift healthcare workers according to their professional category (n = 1,387, AP-HP ALADDIN survey, Paris public hospitals).

| Characteristics | Whole study population | Professional category of NSHW | | | | | p-value[1] |
|---|---|---|---|---|---|---|---|
| | | Nurses | Assistant nurses or technicians | Midwives | Executives | Other categories | |
| | (n = 1,387) | (52.3%) | (38.2%) | (4.2%) | (0.8%) | (4.6%) | |
| | Percent [95% CI♦] or mean (SD) | | | | | | |
| **a. Socio-demographic and economic characteristics** | | | | | | | |
| Female gender | 77.5 [75.1–79.9] | 82.4 | 71.2 | 96.1 | 71.2 | 58.4 | <0.001 |
| **Age—in years** | 39.3 (12.0) | 36.5 (10.7) | 43.4 (10.3) | 33.1 (8.0) | 51.8 (9.5) | 40.3 (11.7) | <0.001 |
| **Matrimonial status** | | | | | | | <0.001 |
| • single | 36.6 [34.0–39.2] | 43.2 | 29.4 | 29.2 | 13.1 | 32.8 | |
| • in cohabitation | 20.8 [18.6–23.0] | 21.1 | 18.9 | 35.3 | 16.6 | 20.9 | |
| • in civil partnership or married | 33.4 [30.8–35.9] | 28.8 | 39.7 | 35.6 | 43.8 | 29.4 | |
| • widow or widower | 9.2 [7.6–10.9] | 7.0 | 12.1 | 0 | 26.5 | 16.9 | |
| **Has children** | | | | | | | <0.001 |
| • no | 49.8 [47.1–52.6] | 60.9 | 33.3 | 67.8 | 24.9 | 49.9 | |
| • yes, and at least one lives at home | 42.4 [39.7–45.1] | 36.9 | 52.1 | 32.2 | 49.5 | 32.7 | |
| • yes, but none at home | 7.8 [6.1–9.5] | 2.2 | 14.7 | 0 | 25.6 | 17.4 | |
| **Has partial or complete custody of at least one child** | 40.5 [37.8–43.1] | 36.0 | 47.8 | 32.2 | 47.6 | 36.3 | <0.001 |
| **Perceived financial status** | | | | | | | <0.001 |
| • Feels financially comfortable/it's okay | 40.0 [37.3–42.7] | 44.9 | 28.9 | 84.5 | 62.0 | 32.8 | |
| • Has to be careful | 46.0 [43.3–48.7] | 46.1 | 50.1 | 15.5 | 35.4 | 40.5 | |
| • Faces financial difficulties | 14.0 [11.9–16.0] | 9.0 | 21.0 | 0 | 2.6 | 26.7 | |
| **b. Work-related characteristics** | | | | | | | |
| **Type of position** | | | | | | | <0.001 |
| • Permanent night position | 75.8 [73.3–78.2] | 76.1 | 84.7 | 4.9 | 84.1 | 61.2 | |
| • Replacement ("pool") | 4.3 [3.2–5.4] | 4.4 | 4.9 | 0 | 0 | 3.7 | |
| • Position with day/night alternation | 16.2 [14.1–18.3] | 16.7 | 8.0 | 95.1 | 3.8 | 9 | |
| • New night-shift position during the pandemic | 0.8 [0.3–1.3] | 0.9 | 0.9 | 0 | 0 | 0 | |
| • Other | 2.9 [1.7–4.1] | 2.0 | 1.5 | 0 | 12.1 | 26.2 | |
| **Hospital department** | | | | | | | <0.001 |
| • Pediatric | 15.1 [13.2–17.1] | 16.9 | 14.7 | 4.9 | 1.4 | 10.5 | |
| • Adults | 61.2 [58.5–63.9] | 66.5 | 60.6 | 46.6 | 23.9 | 25.8 | |
| • Several departments* | 23.7 [21.3–26.0] | 16.6 | 24.7 | 48.5 | 74.7 | 63.7 | |
| **Hospital unit** | | | | | | | <0.001 |
| • Surgery | 16.4 [14.3–18.5] | 16.7 | 15.4 | 37.2 | 9.3 | 3.7 | |
| • Geriatrics/Rehabilitation | 8.8 [7.2–10.5] | 6.9 | 14.0 | 0 | 3.3 | 0 | |
| • Internal medicine/Infectiology/Cardiology/Pneumology | 7.3 [6.0–8.6] | 9.1 | 5.8 | 0 | 1.2 | 7.3 | |
| • Neurology/Nephrology/Oncology/Endocrinology | 6.5 [5.2–7.8] | 8.7 | 5.1 | 0 | 2.4 | 0 | |
| • Pediatrics | 15.7 [13.7–17.7] | 17.3 | 15.8 | 4.9 | 1.4 | 10.5 | |
| • Resuscitation | 13.4 [11.7–15.2] | 17.9 | 10.2 | 0 | 0 | 3.6 | |
| • Emergency | 7.2 [5.7–8.7] | 6.5 | 7.4 | 9.4 | 7.6 | 11.1 | |
| • Several units* | 24.5 [22.1–27] | 16.9 | 26.5 | 48.5 | 74.7 | 63.7 | |
| **Seniority as a night-shift worker—in years** | 9.0 (8.5) | 8.4 (8.1) | 9.5 (8.0) | 9.7 (6.1) | 14.3 (10.6) | 7.8 (9.2) | <0.001 |
| **Daily duration of work** | | | | | | | <0.001 |
| • 10 hours | 62.1 [59.5–64.8] | 61.3 | 72.7 | 0 | 80.2 | 38.1 | |
| • 12 hours | 34.0 [31.4–36.6] | 36.2 | 25.1 | 96.6 | 8.0 | 29.9 | |
| • other | 3.9 [2.6–5.2] | 2.5 | 2.3 | 3.4 | 11.8 | 32.0 | |

*(Continued)*

**Table 1.** (Continued)

| Characteristics | Whole study population | Nurses | Assistant nurses or technicians | Midwives | Executives | Other categories | p-value[1] |
|---|---|---|---|---|---|---|---|
| | | | Professional category of NSHW | | | | |
| | (n = 1,387) | (52.3%) | (38.2%) | (4.2%) | (0.8%) | (4.6%) | |
| **Part-time work** | 5.2 [4.0–6.4] | 5.7 | 4.3 | 9.1 | 7.3 | 3.2 | 0.466 |
| **Travel time to work (home-work one-way commute)—**in minutes | 42 (36) | 42 (24) | 48 (36) | 48 (54) | 36 (24) | 48 (24) | **0.035** |
| **c. Health-related characteristics** | | | | | | | |
| **Perceived health** | | | | | | | **0.019** |
| • Bad or very bad | 8.3 [6.7–9.9] | 8.5 | 7.4 | 4.9 | 7.7 | 17.4 | |
| • Fair | 40.5 [37.7–43.2] | 40.2 | 41.8 | 25.1 | 40.9 | 47.0 | |
| • Good or excellent | 51.3 [48.5–54] | 51.3 | 50.8 | 70.1 | 51.4 | 35.6 | |
| **Practice of any physical activity** | 54.2 [51.2–57.1] | 52.5 | 53.4 | 85.5 | 39.2 | 52.8 | **<0.001** |
| **Perception of a change in weight since working at night** (309 missing values) | 68.2 [65.3–71.1] | 68.1 | 69.8 | 45.0 | 68.1 | 70.6 | **0.251** |
| **History of cancer** | 3.8 [2.6–5.1] | 3.5 | 3.4 | 3.4 | 4.6 | 12.5 | **0.026** |
| **History of psychiatric troubles (depression, bipolar disorders, etc.)** | 5.3 [3.9–6.7] | 4.1 | 5.1 | 7.8 | 10.9 | 17.6 | **0.001** |
| **History of sexual or moral harassment at work** | 20.9 [18.5–23.4] | 20.6 | 21.7 | 12.1 | 39.6 | 23.3 | 0.323 |
| **History of SARS-CoV-2 infection** | | | | | | | **0.014** |
| • No | 58.7 [56.0–61.4] | 60.0 | 60.3 | 43.3 | 59.2 | 44.2 | |
| • Yes | 13.6 [11.7–15.5] | 14.1 | 13.3 | 9.6 | 14.0 | 14.5 | |
| • Did not answer | 27.7 [25.2–30.2] | 25.9 | 26.4 | 47.1 | 26.8 | 41.3 | |
| **d. Work-related perceptions** | | | | | | | |
| **Night-shift work is often or always under-estimated by colleagues working during the day[2]** | 64.7 [62.0–67.4] | 67.1 | 69.6 | 28.0 | 73.6 | 29.9 | **<0.001** |
| **Night-shift work is often or always under-estimated by loved ones[2,3]** | 21.9 [19.6–24.2] | 24.3 | 17.6 | 36.8 | 14.0 | 17.4 | **0.003** |
| **Night-shift work is often or always under-estimated by patients[2]** | 18.3 [16.2–20.4] | 20.2 | 16.9 | 12.2 | 18.2 | 13.6 | 0.322 |
| **Day missions are more important than night missions[4]** | 24.0 [21.7–26.3] | 23.9 | 25.2 | 12.7 | 17.0 | 27.6 | 0.271 |
| **Day workload is higher than night workload[4]** | 38.5 [35.8–41.2] | 42.7 | 33.7 | 18.7 | 17.7 | 52.8 | **<0.001** |
| **Work rhythm is a source of tension with partner or children** (30.9% in the "not concerned" category) | 47.2 [43.1–51.3] | 56.8 | 33.1 | 79.5 | 32.2 | 54.0 | **<0.001** |
| **Work rhythm is a source of tension with friends** | 20.1 [17.9–22.4] | 24.3 | 11.6 | 34.4 | 27.0 | 28.4 | **<0.001** |
| **Feels more irritable since works at night** | 43.6 [40.7–46.5] | 48.7 | 33.6 | 72.8 | 30.1 | 40.3 | **<0.001** |
| **e. Changes in work organization since the beginning of the COVID-19 pandemic** | | | | | | | |
| **No change at all** | 36.8 [34.1–39.5] | 32.5 | 42.8 | 25.4 | 23.2 | 49.8 | **<0.001** |
| **Change of department** | 25.8 [23.5–28.2] | 27.8 | 27.3 | 2.8 | 14.3 | 14.0 | **<0.001** |
| **Change of ward (part of department)** | 29.8 [27.3–32.2] | 33.2 | 31.1 | 0 | 13.6 | 10.8 | **<0.001** |
| **Increase of the no. of working hours** | 37.2 [34.6–39.9] | 39.3 | 29.0 | 71.8 | 65.5 | 45.5 | **<0.001** |
| **Switch to night-shift work** | 5.2 [3.9–6.6] | 5.0 | 4.4 | 4.9 | 5.6 | 15.9 | **0.005** |
| **Change of activity to manage COVID patients** | 19.0 [16.9–21.1] | 23.7 | 15.2 | 14.4 | 8.1 | 3.9 | **<0.001** |
| **f. COVID-related items** | | | | | | | |
| **Satisfied of the information on COVID received from the employer[5]** | 31.5 [29.0–34.0] | 30.2 | 32.2 | 34.1 | 54.1 | 34.5 | 0.440 |
| **Feels vulnerable to COVID-19 because of professional activity[6]** | 77.8 [75.3–80.3] | 80.4 | 73.7 | 89.4 | 63.0 | 72.9 | **0.015** |
| **Fears to get the COVID-19 at work** | 65.5 [62.7–68.3] | 64.2 | 67.0 | 73.7 | 44.3 | 65.0 | 0.368 |
| **Fears to transmit the COVID-19 to close relatives** | 90.6 [88.9–92.4] | 90.4 | 90.5 | 96.5 | 79.3 | 91.3 | 0.477 |

(Continued)

**Table 1.** (Continued)

| Characteristics | Whole study population | Nurses | Assistant nurses or technicians | Midwives | Executives | Other categories | p-value¹ |
|---|---|---|---|---|---|---|---|
| | | | Professional category of NSHW | | | | |
| | (n = 1,387) | (52.3%) | (38.2%) | (4.2%) | (0.8%) | (4.6%) | |
| Has received psychological support from close relatives during the previous two weeks | 7.0 [5.4–8.5] | 7.6 | 7.4 | 0 | 2.0 | 4.3 | 0.286 |
| Has received psychological support from a professional during the previous two weeks | 8.4 [6.7–10.1] | 8.4 | 9.6 | 0 | 3.3 | 8.1 | 0.236 |
| Felt valued by the general population as a NSHW during the pandemic | 62.9 [60.1–65.8] | 65.0 | 59.2 | 76.9 | 63.0 | 55.8 | 0.067 |
| Is confident in the health authorities to manage the crisis⁶ | 19.6 [17.3–22.0] | 18.6 | 20.4 | 23.1 | 43.1 | 17.7 | 0.346 |
| Faced difficulties in applying protective measures against COVID⁶ | 59.7 [56.8–62.6] | 59.2 | 58.9 | 69.1 | 44.9 | 65.9 | 0.454 |
| Considers protective measures inadequate⁶ | 27.6 [24.9–30.2] | 27.6 | 28.5 | 23.6 | 10.9 | 25.7 | 0.727 |
| Faced difficulties in getting screened for SARS-CoV-2 infection⁶ | 58.4 [55.5–61.4] | 57.8 | 62.3 | 56.9 | 37.3 | 38.8 | **0.013** |

CI = confidence interval; NSHW = night-shift healthcare workers; SD = standard deviation.

♦ For the purpose of readability of the table, 95% confidence intervals are only presented for the characteristics of the whole study population.

* Concerns healthcare workers assigned to different departments or units.

¹ Comparison of characteristics between the five professional categories of NSHW (Chi-square tests for categorical variables, Wald test for continuous variables).

² The other possible answers to this item of the questionnaire included "never", "rarely", and "from time to time".

³ Loved ones included partner, family, and friends.

⁴ "I totally agree" or "I agree" (*versus* "I totally disagree" or "I disagree").

⁵ "The information on protective measures against COVID that I received from my employer were sufficient and complete."

⁶ "I totally agree" or "I agree" (*versus* "I totally disagree", "I disagree", or "no interest").

## Distribution of QWL scores

Median [interquartile range, IQR] WRQoL full-scale score was 71 [63–78] in the whole study population (Fig 1). Its distribution was significantly different between professional categories, with the highest score observed amongst executives, and the lowest amongst nurses (in mean (SD): 73 (5.8) *versus* 69.6 (10.6), p = 0.001) (Table 2). The distributions of scores for the six dimensions of QWL are also presented in Table 2. Except for general well-being, in addition to job and career satisfaction, significant differences—globally below 1 or 2 points in median, with a maximum of 3 points—were observed between the QWL scores for the different professional categories. Midwives had the lowest QWL scores for home-work interface, working conditions, and stress at work (meaning impaired QWL for these three dimensions) along with the highest QWL score (meaning better QWL) for control at work. Executives presented higher scores in the "Home-work interface" and "Job and career satisfaction" dimensions of QWL, compared with the other professional categories.

## Determinants of QWL

In the multivariable QWL model, satisfaction on the information on COVID received from the employer and feeling valued by the general population as a NSHW during the pandemic were identified as independent correlates of higher full-scale WRQoL score (Table 3), after adjustment for socio-economic characteristics (matrimonial status, professional category, financial difficulties, hospital unit of assignment), perceived health, history of harassment at work, and physical activity.

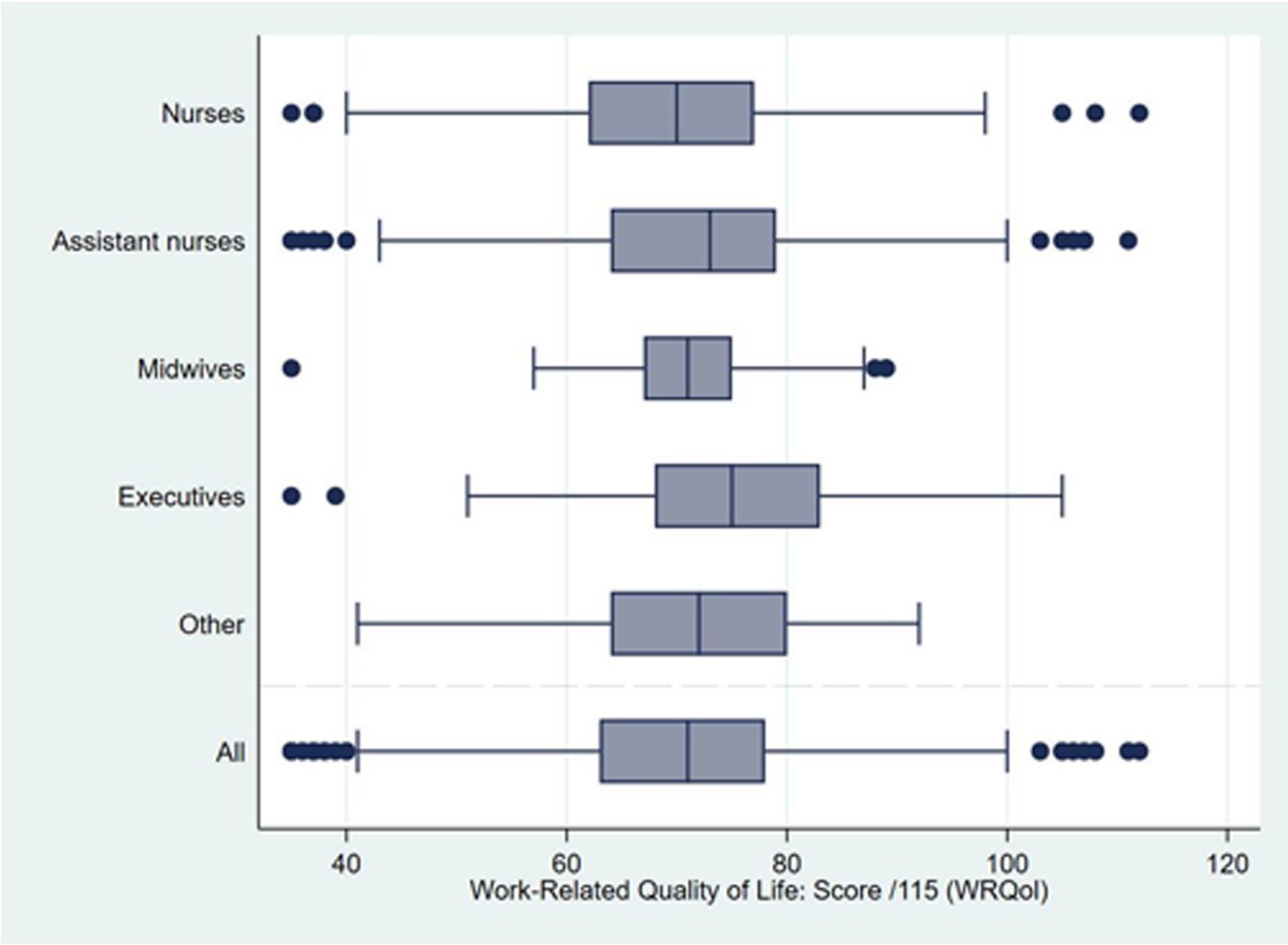

**Fig 1. Boxplots of quality of working life scores among night-shift healthcare workers according to their professional category (n = 1,387, AP-HP ALADDIN survey, Paris public hospitals).** The boxplots present median values and interquartile ranges (box) for the full-scale WRQoL score (range 0 to 115). Lines (whiskers) include all points within 1.5 interquartile range of the nearest quartile. Higher score values denote better QWL.

By contrast, a new night-shift position during the pandemic, under-estimation of night-shift work by colleagues working during the day, work rhythm as a source of tension with friends, feeling more irritable since working at night, considering protective measures against the COVID-19 inadequate, and having faced difficulties in getting screened for SARS-CoV-2 infection were all independent correlates of lower full-scale WRQoL score.

## Discussion

This representative survey offers a comprehensive picture of perceived quality of life at work among NSHW in Paris hospitals shortly after the first wave of the COVID-19 pandemic. After adjustment for socio-demographic, professional, and health-related characteristics, both social and professional recognition of night-shift work appeared as key determinants of QWL in this population. By contrast, lack of or insufficient access to screening, information, and protective measures significantly impaired QWL.

These findings highlight the impact on QWL of the difficulties faced by hospital teams to organize the chain of information and to provide safety equipment to all caregivers during the

**Table 2.** Mean quality of working life scores among night-shift healthcare workers according to their professional category (n = 1,387, AP-HP ALADDIN survey, Paris public hospitals).

| Scores calculated from the WRQoL scale[1] [35] (range) | Whole study population (n = 1,387) | Professional category of NSHW | | | | | p-value[2] |
|---|---|---|---|---|---|---|---|
| | | Nurses (52.3%) | Assistant nurses or technicians (38.2%) | Midwives (4.2%) | Executives (0.8%) | Other categories (4.6%) | |
| | | mean (SD) | | | | | |
| **GLOBAL WORK-RELATED QUALITY OF LIFE SCORE** | | | | | | | |
| **Full-scale WRQoL score** (0 to 115) | 70.5 (12.0) | 69.6 (10.6) | 71.7 (14.2) | 70.2 (15.7) | 73.0 (5.8) | 70.9 (20.1) | **0.001** |
| **SCORES ASSOCIATED WITH THE SIX DIMENSIONS OF WORK-RELATED QUALITY OF LIFE** | | | | | | | |
| **General well-being (GWB)** (0 to 30) | 19.3 (4.2) | 19.2 (3.8) | 19.5 (4.9) | 19.3 (5.3) | 19.2 (1.9) | 19.5 (7.1) | 0.844 |
| **Home-work interface (HWI)** (0 to 15) | 9.2 (2.1) | 9 (1.9) | 9.4 (2.4) | 8.4 (2.9) | 9.6 (1.0) | 9.6 (2.7) | **<0.001** |
| **Job and career satisfaction (JCS)** (0 to 30) | 18.8 (3.8) | 18.7 (3.2) | 18.9 (4.5) | 19.4 (6.2) | 20.4 (1.8) | 18.6 (7.2) | 0.060 |
| **Control at work (CAW)** (0 to 15) | 9.3 (2.6) | 9.2 (2.3) | 9.4 (3.0) | 10.3 (3.5) | 9.9 (1.1) | 8.7 (4.7) | **0.025** |
| **Working conditions (WCS)** (0 to 15) | 8.2 (2.6) | 8.0 (2.4) | 8.5 (2.9) | 7.7 (3.1) | 8.8 (1.2) | 8.7 (3.9) | **0.021** |
| **Stress at work (SAW)** (0 to 10) | 5.7 (1.9) | 5.6 (1.7) | 6.0 (2.1) | 5.1 (2.4) | 5.2 (0.9) | 5.9 (3.0) | **0.002** |

NSHW = night-shift healthcare workers; IQR = interquartile range; SD = standard deviation; WRQoL = work-related quality of life.

[1] For each score, higher values denote better quality of working life.

[2] Comparison of mean scores between the five professional categories of NSHW (Wald test).

first wave of COVID-19, despite their preparedness and training for emergency situations. Indeed, this unexpected global health crisis caused by a previously unknown virus has deeply challenged healthcare workers' adaptability [37], and has stressed the need to update safety guidelines to protect and prevent infection in hospital workers [38]. Another study in the COVID-19 context underlined that coping strategies could influence healthcare workers' well-being and QWL [39]. Findings from ALADDIN also highlight the importance of recognizing the contribution of all healthcare workers [37]. Previous work showed that emphasizing the value of healthcare workers' role was essential to motivate them and to increase their willing-ness to work during public health emergency situations [40]. Professional recognition also includes feeling supported by peers. In the ALADDIN survey, 64.7% of NSHW reported that night-shift work is often or always under-estimated by colleagues working during the day, and this perceived stigma had a significant detrimental effect on QWL. These findings highlight the need to develop interventions to improve communication, sharing of experiences, and support between day-shift and night-shift hospital healthcare workers. Such interventions can reinforce the sense of community among healthcare workers, and have the potential to improve NSHW's experience in the workplace. In addition, findings confirm that healthcare workers' perception of their public image can influence their QWL [41].

Results from the univariable analyses confirm the detrimental effect on QWL of self-per-ceived vulnerability to COVID-19 and fear of transmitting the infection to close relatives. Pre-vious research has also shown a negative psychological impact of these two factors among healthcare workers in France [42]. Interestingly, in ALADDIN, these factors were not identi-fied as independent correlates of QWL in the final multivariable model, maybe because of their correlation with other COVID-related variables such as difficulties to get screened and perceived inadequate and insufficient protective measures. In the same way, changes in work organization since the beginning of the pandemic did not remain in the model after multivari-able adjustment.

**Table 3. Factors associated with quality of working life among night-shift healthcare workers: Linear regression models with full-scale WRQoL score as the outcome (n = 1,387, ALADDIN survey, Paris public hospitals).**

| Characteristics | Univariable models | | | Multivariable model (n = 1,124) | | |
|---|---|---|---|---|---|---|
| | Coefficient | [95% CI] | *p-value* | Adjusted coefficient | [95% CI] | *p-value* |
| **SOCIO-DEMOGRAPHIC AND ECONOMIC CHARACTERISTICS** | | | | | | |
| **Matrimonial status** | | | | | | |
| • single | ref | ref | | ref | ref | |
| • in cohabitation | 0.51 | [-1.55; 2.56] | 0.627 | 1.03 | [-0.68; 2.74] | 0.238 |
| • in civil partnership or married | 1.17 | [-0.54; 2.87] | 0.181 | 0.18 | [-1.32; 1.68] | 0.814 |
| • widow or widower | 3.37 | [0.97; 5.76] | 0.006 | 2.45 | [0.09; 4.81] | **0.042** |
| **Perceived financial status** | | | | | | |
| • Feels financially comfortable/it's okay | ref | ref | | ref | ref | |
| • Has to be careful | -3.58 | [-5.09; -2.07] | <0.001 | -2.07 | [-3.49; -0.65] | **0.004** |
| • Faces financial difficulties | -6.65 | [-9.05; -4.26] | <0.001 | -4.87 | [-7.17; -2.58] | <**0.001** |
| **WORK-RELATED CHARACTERISTICS** | | | | | | |
| **Professional category** | | | | | | |
| • Nurses | ref | ref | | ref | ref | |
| • Assistant nurses or technicians | 2.08 | [0.52; 3.63] | 0.009 | 1.93 | [0.50; 3.37] | **0.008** |
| • Midwives | 0.53 | [-3.99; 5.05] | 0.819 | 0.13 | [-4.06; 4.33] | 0.950 |
| • Executives | 3.38 | [-0.21; 6.98] | 0.065 | 1.68 | [-1.18; 4.54] | 0.250 |
| • Other categories | 1.30 | [-3.68; 6.27] | 0.609 | 0.62 | [-3.36; 4.60] | 0.760 |
| **Type of position** | | | | | | |
| • Permanent night position | ref | ref | | | | |
| • Replacement ("pool") | -2.70 | [-7.01; 1.62] | 0.220 | -2.33 | [-5.28; 0.63] | 0.123 |
| • Position with day/night alternation | -0.32 | [-2.02; 1.38] | 0.709 | -1.87 | [-4.10; 0.36] | 0.100 |
| • New night-shift position during the COVID pandemic | -13.43 | [-25.37; -1.50] | 0.027 | -12.56 | [-23.81; -1.31] | **0.029** |
| • Other | -0.02 | [-5.07; 5.04] | 0.995 | -0.37 | [-4.76; 4.02] | 0.869 |
| **Hospital unit** | | | | | | |
| • Surgery | ref | ref | | ref | ref | |
| • Geriatrics/Rehabilitation | 4.96 | [1.79; 8.12] | 0.002 | 2.92 | [0.22; 5.63] | **0.034** |
| • Internal medicine/Infectiology/Cardiology/Pneumology | 0.82 | [-2.27; 3.92] | 0.602 | 1.03 | [-1.71; 3.76] | 0.461 |
| • Neurology/Nephrology/Oncology/Endocrinology | 2.00 | [-1.22; 5.21] | 0.224 | 1.80 | [-1.07; 4.67] | 0.218 |
| • Pediatrics | 2.89 | [0.20; 5.58] | 0.035 | 1.33 | [-0.99; 3.65] | 0.262 |
| • Resuscitation | 2.24 | [0.04; 4.43] | 0.046 | 3.00 | [0.89; 5.11] | **0.005** |
| • Emergency | 1.09 | [-2.14;4.32] | 0.508 | 1.48 | [-1.24; 4.21] | 0.286 |
| • Several units | 1.87 | [-0.21; 3.95] | 0.077 | 1.32 | [-0.66; 3.31] | 0.191 |
| **HEALTH-RELATED CHARACTERISTICS** | | | | | | |
| **Perceived health** | | | | | | |
| • Bad or very bad | ref | ref | | ref | ref | |
| • Fair | 7.39 | [4.20; 10.57] | <0.001 | 4.98 | [2.20; 7.76] | <**0.001** |
| • Good or excellent | 13.56 | [10.4; 16.72] | <0.001 | 8.80 | [5.99; 11.61] | <**0.001** |
| **Practice of any physical activity** | 3.54 | [2.05; 5.03] | <0.001 | 1.33 | [0.08; 2.57] | **0.037** |
| **History of sexual or moral harassment at work** | -5.36 | [-7.36; -3.35] | <0.001 | -3.70 | [-5.42; -1.98] | <**0.001** |
| **WORK-RELATED PERCEPTIONS** | | | | | | |
| **Night-shift work is often or always under-estimated by colleagues working during the day[1]** | -3.95 | [-5.57; -2.32] | <0.001 | -3.54 | [-5.01; -2.08] | <**0.001** |
| **Work rhythm is a source of tension with friends** | -7.11 | [-9.08; -5.14] | <0.001 | -2.28 | [-4.14; -0.42] | **0.016** |
| **Feels more irritable since works at night** | -5.46 | [-6.92; -3.99] | <0.001 | -2.96 | [-4.34; -1.58] | <**0.001** |

*(Continued)*

**Table 3.** (Continued)

| Characteristics | Univariable models | | | Multivariable model (n = 1,124) | | |
|---|---|---|---|---|---|---|
| | Coefficient | [95% CI] | p-value | Adjusted coefficient | [95% CI] | p-value |
| WORK ORGANIZATION: CHANGES SINCE THE BEGINNING OF THE COVID PANDEMIC | | | | | | |
| COVID-RELATED ITEMS | | | | | | |
| Satisfied of the information on COVID received from the employer[4] | 7.36 | [5.86; 8.85] | <0.001 | 4.67 | [3.26; 6.07] | **<0.001** |
| Felt valued by the general population as a NSHW during the pandemic | 3.12 | [1.52; 4.71] | <0.001 | 1.41 | [0.13; 2.70] | **0.031** |
| Considers protective measures inadequate[5] | -5.28 | [-6.98; -3.57] | <0.001 | -2.09 | [-3.52; -0.65] | **0.004** |
| Faced difficulties in getting screened for SARS-CoV-2 infection[5] | -4.38 | [-5.89; -2.88] | <0.001 | -2.95 | [-4.25; -1.65] | **<0.001** |

CI = confidence interval; WRQoL = work-related quality of life.

♦ This variable was not entered in the multivariable analysis due to a high rate of NSHW in the "not concerned" category.

[1] The other possible answers to this item of the questionnaire included "never", "rarely", and "from time to time".

[2] Loved ones included partner, family, and friends.

[3] "I totally agree" or "I agree" (*versus* "I totally disagree" or "I disagree").

[4] "The information on protective measures against COVID that I received from my employer were sufficient and complete."

[5] "I totally agree" or "I agree" (*versus* "I totally disagree", "I disagree", or "no interest").

There is a lack of published studies on QWL conducted among healthcare workers, especially in France. We identified only one recent survey, also based on the WRQoL scale [43]. QWL level observed in ALADDIN was lower than that found in this recent survey, conducted among 2,040 French anesthesiologists (median [IQR] WRQoL full-scale score: 71 [63–78] *versus* 77 [66–85]) [43]. This difference is likely to be related to the study period, as the latter survey was performed before the beginning of the COVID-19 pandemic (January to June 2019). It may also be related to the diversity of professional categories participating in ALADDIN, presenting different levels of QWL.

Compared with the WRQoL scale's norms, the median QWL score in ALADDIN corresponds to a relatively low level of QWL. However, these norms refer to the UK National Health Services [36], and may not be adapted to the French context because of differences between countries in the organization and functioning of healthcare services. Cultural specificities may also play a role, as shown in other research areas such as perception of happiness [44]. These specificities may be linked to differences in people's work-related representations and expectancies. Environmental factors such as the socio-political context in different countries may make international comparisons even more difficult.

Findings from ALADDIN showed statistically significant differences in QWL between professional categories. These differences were however of modest magnitude and did not exceed 3 points in QWL scores. Further research is needed to determine if such a magnitude exceeds the minimum important difference for the WRQoL scale. Executives showed both the best overall QWL and higher scores in the "Home-work interface" and "Job and career satisfaction" dimensions of QWL, compared with that of other professional categories. Along with older age, correlated with less domestic responsibilities related to child care, a longer experience of night-shift work may explain the greater ability of executives to find the right balance between their professional and personal lives. By contrast, executives (together with midwives) presented a low score of QWL in the "Stress at work" dimension, revealing higher levels of stress than other professional categories. Interestingly, midwives reported the lowest QWL related to working conditions. Further research should thus be performed to identify midwives' specific needs and expectations to both improve their QWL and prevent psychosocial risks [45]. Of

note, the number of years in night-shift work (variable "seniority as a night-shift worker (in years)") was not significantly associated with overall QWL, despite its heterogeneity in our study sample. We hypothesize that seniority may influence one's night-shift work experience in different ways. For instance, workers with more night-shift work experience may better cope with stress than those with less experience. By contrast, the latter may have been less exposed to changes in the circadian rhythm, resulting in better perceived health.

The ALADDIN survey has several strengths. First, its representativeness regarding sex, age, and professional categories allows presenting a snapshot of QWL among all NSHW working in Paris public hospitals. Second, the choice of the study period, which directly followed the first wave of COVID-19 in France (March to June 2020), is adequate to assess NSHW's perceptions during the pandemic. Indeed, once their work overload started to decline after the peak of the crisis, NSHW were more prone to both share their feelings and experiences, and to assess the repercussions of the pandemic on their QWL. Lastly, the ALADDIN survey explores a large panel of potential correlates of QWL, using a standard scale (WRQoL).

However, the survey is limited by its cross-sectional design. Further research is therefore needed to assess longitudinal changes in QWL among NSHW throughout the pandemic, and in the long term. Another limitation of our study is the lack of comparative data among day-shift hospital workers. Such data would have helped distinguish between the effects of shift-work by itself on QWL and those related to coping with the pandemic. Future surveys should include both populations of hospital workers. Of note, external factors such as the time of day the questionnaire was completed may have influenced NSHW's answers (notably due to fatigue). This type of bias, inherent to self-reported data, is difficult to take into account in the analyses. Indeed, a potential "time of the day" effect depends on many unmeasured factors, including NSHW's number of hours worked before completing the questionnaire, their workload, and inter-individual variations in the internal clock (some individuals feel awake late at night, whilst others are sleepy).

In France, there is a growing interest for healthcare professionals' quality of life at work, with a national strategy for the improvement of QWL ("Caring the caregivers"), aiming notably at improving work environment and work conditions, informing managers about QWL-related issues and psychosocial risks, and supporting them in the adoption of better work methods [46]. In line with this strategy, a national observatory was created in 2018 to monitor QWL among healthcare and medico-social workers. The COVID-19 pandemic has further stressed the need to document QWL in healthcare services, and to identify its determinants during and after such sanitary crises [47]. Findings from the AP-HP ALADDIN survey contribute to increase the body of knowledge about these key issues, which are central to set up efficient strategies to reinforce healthcare systems. Such strategies should include interventions aiming to improve recognition, reduce stigma related to night-shift work, and to improve information and communication between the different groups of healthcare workers.

To conclude, in this representative survey, insufficient access to screening, information, and protective measures impaired QWL of NSHW after the first wave of COVID-19 in Paris public hospitals. Social and professional recognition of night-shift work appear as key determinants of QWL in this population. Further research is needed to monitor longitudinal changes in QWL of NSHW during and after the different waves of the COVID-19 pandemic.

## Acknowledgments

Our thanks to all participants in the AP-HP ALADDIN survey. We also thank Isabelle Chavignaud (FIDES mission) for her help in developing the survey, all medicine students who actively helped us recruit participants, and the AP-HP communication department for their

help in promoting the survey. Finally, our thanks to Carter Brown and Lauren Perieres for the English revision and copyediting of the manuscript.

## Author Contributions

**Conceptualization:** Martin Duracinsky, Lorraine Cousin, Véronique Mahé, Olivia Rousset-Torrente.

**Data curation:** Martin Duracinsky, Lorraine Cousin, Vincent Di Beo, Olivia Rousset-Torrente.

**Formal analysis:** Vincent Di Beo.

**Funding acquisition:** Martin Duracinsky, Lorraine Cousin, Olivia Rousset-Torrente, Patrizia Carrieri, Olivier Chassany.

**Investigation:** Martin Duracinsky, Fabienne Marcellin, Lorraine Cousin, Vincent Di Beo, Véronique Mahé, Patrizia Carrieri, Olivier Chassany.

**Methodology:** Martin Duracinsky, Fabienne Marcellin, Lorraine Cousin, Vincent Di Beo, Véronique Mahé, Patrizia Carrieri, Olivier Chassany.

**Project administration:** Martin Duracinsky, Lorraine Cousin, Olivia Rousset-Torrente.

**Resources:** Martin Duracinsky, Lorraine Cousin, Olivia Rousset-Torrente.

**Supervision:** Martin Duracinsky, Patrizia Carrieri, Olivier Chassany.

**Writing – original draft:** Martin Duracinsky, Fabienne Marcellin, Lorraine Cousin.

**Writing – review & editing:** Martin Duracinsky, Fabienne Marcellin, Lorraine Cousin, Vincent Di Beo, Véronique Mahé, Olivia Rousset-Torrente, Patrizia Carrieri, Olivier Chassany.

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
