## [Decision Letter · Decision Letter 0]

2 Dec 2021

PONE-D-21-27355Social and professional
recognition are key determinants of quality of life at work among night-shift
healthcare workers in Paris public hospitals (AP-HP ALADDIN COVID-19
survey)PLOS ONE

Dear Dr. Marcellin,

Thank you for submitting your manuscript to PLOS ONE. After careful consideration, we
feel that it has merit but does not fully meet PLOS ONE’s publication criteria as it
currently stands. Therefore, we invite you to submit a revised version of the
manuscript that addresses the points raised during the review
process.

Please submit your revised manuscript by Jan 16 2022 11:59PM. If you will need more
time than this to complete your revisions, please reply to this message or contact
the journal office at plosone@plos.org. When
you're ready to submit your revision, log on to https://www.editorialmanager.com/pone/ and select the 'Submissions
Needing Revision' folder to locate your manuscript file.

Please include the following items when submitting your revised
manuscript:A rebuttal letter that responds to each point raised by the academic
editor and reviewer(s). You should upload this letter as a separate file
labeled 'Response to Reviewers'.A marked-up copy of your manuscript that highlights changes made to the
original version. You should upload this as a separate file labeled
'Revised Manuscript with Track Changes'.An unmarked version of your revised paper without tracked changes. You
should upload this as a separate file labeled 'Manuscript'.

If you would like to make changes to your financial disclosure, please include your
updated statement in your cover letter. Guidelines for resubmitting your figure
files are available below the reviewer comments at the end of this letter.

We look forward to receiving your revised manuscript.

Kind regards,

Florian Fischer

Academic Editor

PLOS ONE

Journal Requirements:

Reviewers' comments:

Reviewer's Responses to Questions

**Comments to the Author**

1. Is the manuscript technically sound, and do the data support the conclusions?

Reviewer #1: Partly

Reviewer #2: No

2. Has the statistical analysis been performed
appropriately and rigorously? 

Reviewer #1: No

Reviewer #2: Yes

3. Have the authors made all data underlying the
findings in their manuscript fully available?

Reviewer #1: No

Reviewer #2: Yes

4. Is the manuscript presented in an intelligible
fashion and written in standard English?

Reviewer #1: Yes

Reviewer #2: No

5. Review Comments to the Author

Reviewer #1: Overview:

The authors evaluated a questionnaire of determinants of quality of life at work
among night-shift healthcare workers in Paris public hospitals. The central idea of
evaluating the quality of life of night workers during the first wave of COVID is
very interesting. Both for the impact on the biological rhythms of night work and
for the challenges of working on the front lines during this period of pandemic. The
idea is therefore very commendable.

However, the great challenge of this work should be the demonstration with subjective
and perhaps objective data, which are the factors that negatively impact the quality
of life inherent to night work (eg: factor "x" = night work) plus the factors that
coping with the pandemic itself, regardless of the work shift, is also negatively
affected (eg factor "y" = COVID pandemic). This distinction is not clearly described
and analyzed in this manuscript.

An example: they could also have analyzed the day shift workers in the same period
and compared the data. We would have a more accurate view of the “pandemic x work
shift” effect.

Minor concerns

The worker's experience in the night shift should be considered and analyzed
separately. Workers with less than three weeks on the night scale will not show the
same effects perceived as "negative" by workers with one year on the night scale.
Likewise, workers with “a lot” of time on the night shift schedule could already be
more physically and psychologically tolerant to typical circadian changes and, in
many cases, also feeling less of these effects.

Major concerns

The time of day when the questionnaires were answered also influence the results.
From a chronobiological point of view, a night shift worker who answered an online
questionnaire at 5 pm, after a period of rest, will likely have a different score if
the same worker answers the same questionnaire at 5 am, after your work’s turn. This
should be described and considered in this manuscript.

All results were described in tables only. Graphic resources with joint presentation
as well as correlation data described separately would be essential. All data in
tables is overly descriptive and difficult to visualize. It can be confusing for the
reader to try to compare data presented at the beginning with data at the end of the
table.

History of psychiatric troubles (depression, bipolar disorders) seems to be a factor
that directly contributes to the subjective answers (questionnaires) related to
quality of life. In this case, it would be interesting for the authors to also
present this data separately, considering the presence or absence of this factor in
the other results.

Discussion:

The title of the manuscript focuses on the results of workers' recognition during the
pandemic as keys to improving quality of life. However, this data (recognition) was
mentioned only 4 times in the entire manuscript, and it was not properly
discussed.

The feeling of being vulnerable to COVID infection, as well as the fear of
transmitting the disease to family members, can be crucial factors in these workers'
negative quality of life scores. Still, the sudden change in the organization of
work seems to be another impacting factor on quality of life, especially for nurses.
These data should be discussed with other results already presented in the
literature, when one wants to compare the effects of the pandemic on these
professionals, regardless of the work shift.

It would be essential, at least in the discussion, to describe what the literature
presented as "quality of life", with these same analysis tools, of workers before
the pandemic (control). The work of Gafsou B, et. al, 2021 could have been more
discussed. Again, the data were only presented descriptively. What was expected is
an intense discussion of each variable.

The authors commented ,"Cultural specificities may also play a role, as shown in
other research areas such as perception of happiness" and that the WRQoL scale's
norms is a British instrument (UK) and may not be It is possible to compare with the
French context, due to cultural differences between countries. However, to assess
the QWL the WRQol scale was used. In other hand, was used this same instrument for
the population of French workers. Wouldn't that be contradictory and difficult to
trust the data presented?

Finally, in order to affirm the effects of the pandemic on shift workers, in addition
to the cross-sectional portrait, the authors are enhanced to explore the
longitudinal analysis of the data, so that we can comprehensively understand the
impacts of this period on this specific population of workers.

Reviewer #2: This paper aimed to document the determinants of quality of working life
(QWL) among night-shift healthcare workers (NSHW) in Paris public hospitals after
the first-wave of the COVID-19 pandemic. However, there are few issues that were
unclear to me and need further clarification, as below:

1. The use of “several department”, “ several unit” are unclear to me. Suggestion for
the authors is to explain these variable under footnote.

2. AP-HP is not spelled out in abstract.

3. The significance of this study is not clearly provided.

4. No information is given on the first wave of pandemic? Why the assessment was done
after first wave? What is the significance of having this study after first
wave?

5. “The ALADDIN cross-sectional survey (15 June 2020 to 15 September 2020) 76 was
conducted among NSHW in the 39 hospitals of the Assistance Publique - Hôpitaux de
Paris (AP-HP).” How the hospitals are selected? And any inclusion/ exclusion
criterias for the selection?

6. What is the response rate? This may lead to information bias when the response
rate is too low.

7. Few of the variables, i.e., “physical activity”, “Change in weight since works at
night”, “Travel time to work” are unclear to me. What are the operational
definitions for these variables. Table 3: 0.7 hours in the travel time means?

8. Page 29, Line 238: “social and professional recognition ” is unclear to me. Please
explain.

9. Table 1 is poorly presented.

10. Page 30, Line 283: “The lack of reference values for the WRQoL scale in France
also limits the discussion of results.”. What is the importance of having reference
values in this study?

11. Page 30, Line 263: “These differences were however of modest magnitude and did
not exceed 3 points in QWL scores.” Is 3 points as a cutoff value to determine the
difference between groups? Please provide references.

12. Page 31, Line 290: “Findings from the AP-HP ALADDIN survey contribute to increase
the body of knowledge about these key issues, which are central to set up efficient
strategies to reinforce healthcare systems.” What are the examples of strategies
that could be recommended by the authors based on the findings of this study.

13. The findings of Table 2 is poorly discussed.

6. PLOS authors have the option to publish the peer
review history of their article (what does this mean?). If published, this will
include your full peer review and any attached files.

If you choose “no”, your identity will remain anonymous but your review may still be
made public.

**Do you want your identity to be public for this peer review?** For
information about this choice, including consent withdrawal, please see our
Privacy Policy.

Reviewer #1: No

Reviewer #2: No

---

## [Author Response · Author response to Decision Letter 0]

11 Feb 2022

Fabienne MARCELLIN 

INSERM UMR1252- SESSTIM

Faculté de Médecine

3e étage - Aile bleue

27 boulevard Jean Moulin

13385 Marseille cedex 5 FRANCE

e-mail: fabienne.marcellin@inserm.fr

Marseille, 11 February 2022

To Prof. Florian Fischer, Academic Editor, PLoS One

Dear Editor,

Thank you very much for giving us the opportunity to submit a revised version of our
manuscript entitled “Social and professional recognition are key determinants of
quality of life at work among night-shift healthcare workers in Paris public
hospitals (AP-HP ALADDIN COVID-19 survey)” to PLoS One.

Below, please find detailed responses to the reviewers’ comments. 

As requested, we have attached a Microsoft Word version of the revised manuscript
that highlights changes made to the original version, and a final unmarked version
of the revised manuscript. 

In addition, as requested in the second message received from Dr Bendaña (9 February
2022), we have updated our Data Availability Statement.

We hope that this revised version will meet the criteria for publication in PLoS One,
and we remain available to make any changes which could further improve our
manuscript.

Best regards,

Fabienne MARCELLIN, corresponding author

Reviewers' comments: 

Reviewer's Responses to Questions 

Comments to the Author

1. Is the manuscript technically sound, and do the data support the conclusions? 

Reviewer #1: Partly 

Reviewer #2: No 

We have complemented the discussion of our results, as suggested by the reviewers
(please see answers to reviewers’ comments in point 5).

2. Has the statistical analysis been performed appropriately and rigorously? 

Reviewer #1: No 

Reviewer #2: Yes 

We have performed the complementary statistical analyses asked by the reviewers
(please see answers to reviewers’ comments in point 5).

3. Have the authors made all data underlying the findings in their manuscript fully
available? 

Reviewer #1: No 

Reviewer #2: Yes

A data sharing statement has been added in the revised manuscript, as follows:

“DATA SHARING STATEMENT

Data is available upon request to the scientific committee of the ALADDIN
survey.”

4. Is the manuscript presented in an intelligible fashion and written in standard
English? 

Reviewer #1: Yes 

Reviewer #2: No 

The manuscript has been re-read and copyedited by two native US English speakers.

5. Review Comments to the Author 

Reviewer #1: Overview: 

The authors evaluated a questionnaire of determinants of quality of life at work
among night-shift healthcare workers in Paris public hospitals. The central idea of
evaluating the quality of life of night workers during the first wave of COVID is
very interesting. Both for the impact on the biological rhythms of night work and
for the challenges of working on the front lines during this period of pandemic. The
idea is therefore very commendable. 

We thank the reviewer for this positive feedback.

However, the great challenge of this work should be the demonstration with subjective
and perhaps objective data, which are the factors that negatively impact the quality
of life inherent to night work (eg: factor "x" = night work) plus the factors that
coping with the pandemic itself, regardless of the work shift, is also negatively
affected (eg factor "y" = COVID pandemic). This distinction is not clearly described
and analyzed in this manuscript. 

An example: they could also have analyzed the day shift workers in the same period
and compared the data. We would have a more accurate view of the “pandemic x work
shift” effect. 

We agree with the reviewer concerning the difficulties to distinguish between the
effects of shift-work by itself and those related to coping with the pandemic. Our
model was adjusted for factors related to each of these two domains (characteristics
and organization of work, perceptions, and experience since the beginning of the
pandemic), leading to an estimation of each effect independently of the others.

As noted by the reviewer, it would have been interesting to compare the level and
correlates of quality of life at work between day-shift workers and night-shift
workers. This should be explored in future surveys.

We have noted this as a limitation of our study in the Discussion section of the
revised manuscript, as follows:

“Another limitation of our study is the lack of comparative data among day-shift
hospital workers. Such data would have helped distinguish between the effects of
shift-work by itself on QWL and those related to coping with the pandemic. Future
surveys should include both populations of hospital workers.”

Minor concerns 

The worker's experience in the night shift should be considered and analyzed
separately. Workers with less than three weeks on the night scale will not show the
same effects perceived as "negative" by workers with one year on the night scale.
Likewise, workers with “a lot” of time on the night shift schedule could already be
more physically and psychologically tolerant to typical circadian changes and, in
many cases, also feeling less of these effects. 

This is an interesting point. We have already described the variable “Seniority as a
night-shift worker,” and tested it as a potential correlate of quality of life at
work.

As shown by the univariable analyses, it was not significantly associated with the
full-scale WRQoL score (coefficient [95% confidence interval: 0.01 [-0.09; 0.11],
p=0.811), even if hospital workers’ years of experience with night-shift work varied
greatly (mean (standard deviation) of seniority as a night-shift worker: 9 (8.5)
years). 

We have discussed this point in the revised manuscript, as follows:

“Of note, the number of years in night-shift work (variable “seniority as a
night-shift worker (in years)”) was not significantly associated with overall QWL,
despite its heterogeneity in our study sample. We hypothesize that seniority may
influence one’s night-shift work experience in different ways. For instance, workers
with more night-shift work experience may better cope with stress than those with
less experience. By contrast, the latter may have been less exposed to changes in
the circadian rhythm, resulting in better perceived health.” 

Major concerns 

The time of day when the questionnaires were answered also influence the results.
From a chronobiological point of view, a night shift worker who answered an online
questionnaire at 5 pm, after a period of rest, will likely have a different score if
the same worker answers the same questionnaire at 5 am, after your work’s turn. This
should be described and considered in this manuscript. 

In the analyses, it is difficult to take into account the time of day when the
questionnaire was filled in. Indeed, the study sample includes individuals with
different work schedules (some having permanent night positions, others working
alternatively between day and night shift). In addition, we have not collected
information on the number of hours worked and the global workload of survey
participants prior to completing the questionnaire. A worker completing the
questionnaire at 2 a.m. may feel exhausted from a non-stop 4-hour rush, while
another may be in good shape, if activity is calm in the department.
Inter-individual variations in the internal clock further complexify the
interpretation of a possible “time of the day” effect (some people feeling
comfortable and awake even late at night, and others feeling sleepy at the same
period).

We have added a point in the Discussion section of the revised manuscript, as
follows:

“Of note, external factors such as the time of day the questionnaire was completed
may have influenced NSHW’s answers (notably due to fatigue). This type of bias,
inherent to self-reported data, is difficult to take into account in the analyses.
Indeed, a potential “time of the day” effect depends on many unmeasured factors,
including NSHW’s number of hours worked before completing the questionnaire, their
workload, and inter-individual variations in the internal clock (some individuals
feel awake late at night, whilst others are sleepy).”

All results were described in tables only. Graphic resources with joint presentation
as well as correlation data described separately would be essential. 

We have added a graphical representation of QWL scores using boxplots, as shown
below. This enabled us to alleviate data presented in Table 2 (please see revised
manuscript).

Figure 1 - Boxplots of quality of working life scores among night-shift healthcare
workers according to their professional category (n=1,387, AP-HP ALADDIN survey,
Paris public hospitals)

The boxplots present median values and interquartile ranges (box) for the full-scale
WRQoL score (range 0 to 115). Lines (whiskers) include all points within 1.5
interquartile range of the nearest quartile. Higher score values denote better
QWL.

All data in tables is overly descriptive and difficult to visualize. It can be
confusing for the reader to try to compare data presented at the beginning with data
at the end of the table. 

We agree that the presentation of descriptive statistics in Table 1 may be difficult
to read due to the high number of lines. Consequently, we have split the data into
six different sub-tables, each representing one group of variables (from
“Sociodemographic and economic characteristics” to “COVID-related items”). This will
facilitate the interpretation of the tables, while keeping information on each
variable analyzed available to readers. 

In addition, in Table 3, we have deleted information about variables which did not
remain in the final multivariable model.

We have also changed double-spacing to 1.5-line spacing in all tables. 

History of psychiatric troubles (depression, bipolar disorders) seems to be a factor
that directly contributes to the subjective answers (questionnaires) related to
quality of life. In this case, it would be interesting for the authors to also
present this data separately, considering the presence or absence of this factor in
the other results. 

The sample size of NSHW with a history of psychiatric troubles was too small in our
survey (n=60) to perform a stratified analysis. Indeed, the lack of statistical
power and unbalanced distribution of the variable “history of psychiatric troubles”
(4.3% “yes” versus 95.7% “no” in the dataset of the multivariable model) do not
enable neither the correct identification of QWL correlates nor the comparison of
odds ratios between the two groups.

Discussion: 

The title of the manuscript focuses on the results of workers' recognition during the
pandemic as keys to improving quality of life. However, this data (recognition) was
mentioned only 4 times in the entire manuscript, and it was not properly discussed. 

We have added a description of variables related to social and professional
recognition in the Methods section of the revised manuscript, as follows (Data
collection paragraph):

“NSHW’s perceptions regarding their social and professional recognition were assessed
using items related to under-estimation of night-shift work by colleagues, loved
ones, and patients; perceptions of the importance of night missions and of workload
during night; feeling valued by the general population as a NSHW during the
pandemic. Most of these items were derived from different stigma scales (Brakel WHV
2006, Berger BE et al. 2001, Golay P et al. 2021)”.

We have also enriched the discussion of workers’ recognition during the pandemic, as
follows:

“Professional recognition also includes feeling supported by peers. In the ALADDIN
survey, 64.7% of NSHW reported that night-shift work is often or always
under-estimated by colleagues working during the day, and this perceived stigma had
a significant detrimental effect on QWL. These findings highlight the need to
develop interventions to improve communication, sharing of experiences, and support
between day-shift and night-shift hospital healthcare workers. Such interventions
can reinforce the sense of community among healthcare workers, and have the
potential to improve NSHW’s daily experience in the workplace.”

The feeling of being vulnerable to COVID infection, as well as the fear of
transmitting the disease to family members, can be crucial factors in these workers'
negative quality of life scores. Still, the sudden change in the organization of
work seems to be another impacting factor on quality of life, especially for nurses.
These data should be discussed with other results already presented in the
literature, when one wants to compare the effects of the pandemic on these
professionals, regardless of the work shift. 

We chose to focus our discussion on variables which were significantly associated
with quality of working life in the multivariable model. Fear of getting infected or
to transmit the disease to family members, and changes in work organization have not
been identified as independent correlates of quality of working life in the final
model.

Nevertheless, in the revised Discussion section, we have added information on these
variables, as suggested:

“Results from the univariable analyses confirm the detrimental effect on QWL of
self-perceived vulnerability to COVID-19 and fear of transmitting the infection to
close relatives. Previous research has also shown a negative psychological impact of
these two factors among healthcare workers in France (Chene G et al 2021).
Interestingly, in ALADDIN, these factors were not identified as independent
correlates of QWL in the final multivariable model, maybe because of their
correlation with other COVID-related variables such as difficulties to get screened
and perceived inadequate and insufficient protective measures. In the same way,
changes in work organization since the beginning of the pandemic did not remain in
the QWL model after multivariable adjustment.”

It would be essential, at least in the discussion, to describe what the literature
presented as "quality of life", with these same analysis tools, of workers before
the pandemic (control). The work of Gafsou B, et. al, 2021 could have been more
discussed. Again, the data were only presented descriptively. What was expected is
an intense discussion of each variable. 

To our knowledge, the only previous study which used the WRQoL scale in the French
context was of the one by Gafsou et al, which was conducted before the COVID
pandemic and can thus be used as a reference. 

We have underlined the lack of published data, as follows (Discussion section):

“There is a lack of published studies on QWL conducted among healthcare workers,
especially in France. We identified only one recent survey, which was also based on
the WRQoL scale (Gafsou et al).”

The authors commented,"Cultural specificities may also play a role, as shown in other
research areas such as perception of happiness" and that the WRQoL scale's norms is
a British instrument (UK) and may not be It is possible to compare with the French
context, due to cultural differences between countries. However, to assess the QWL
the WRQol scale was used. In other hand, was used this same instrument for the
population of French workers. Wouldn't that be contradictory and difficult to trust
the data presented?

The use of the WRQoL scale is justified here, as it is a standard and validated
psychometric tool that enables comparisons between past studies conducted in
specific subgroups of French healthcare workers (such as anesthesiologists in the
work by Gafsou et al.) as well as with future studies assessing QWL (in healthcare
contexts or in other professional contexts). 

However, comparing the level of QWL between individuals from different countries
remains difficult because of differences in people’s work-related representations
and expectancies (what we called “cultural specificities”), and differences in
environmental factors such as the social and political context. Of note, UK and
France share common characteristics, as both are Northern, high-resource countries
of the European geographic region.

We have added the following sentences in the discussion of results:

“These specificities may be linked to differences in people’s work-related
representations and expectancies. Environmental factors such as the socio-political
context in different countries may make international comparisons even more
difficult.”

Finally, in order to affirm the effects of the pandemic on shift workers, in addition
to the cross-sectional portrait, the authors are enhanced to explore the
longitudinal analysis of the data, so that we can comprehensively understand the
impacts of this period on this specific population of workers. 

Unfortunately, data collected are cross-sectional. Even if a “longitudinal” dimension
has been explored in certain items of the questionnaire (such as variables related
to changes in work organization since the beginning of the pandemic), future studies
are needed to better understand the impact of the COVID period among NSHW,
especially in the long term.

We have modified one sentence in the limitations section of the revised manuscript as
follows:

“However, the survey is limited by its cross-sectional design. Further research is
therefore needed to assess longitudinal changes in QWL among NSHW throughout the
pandemic, and in the long term.”

Reviewer #2: 

This paper aimed to document the determinants of quality of working life (QWL) among
night-shift healthcare workers (NSHW) in Paris public hospitals after the first-wave
of the COVID-19 pandemic. However, there are few issues that were unclear to me and
need further clarification, as below: 

1. The use of “several department”, “several unit” are unclear to me. Suggestion for
the authors is to explain these variable under footnote. 

The categories “several departments” and “several units” include NSHW assigned to
different departments or units (for instance, people working both in adult and
pediatric departments).

We have added the following footnote in Table 1b:

“* Concerns healthcare workers assigned to different departments or units.” 

2. AP-HP is not spelled out in abstract. 

This has been corrected.

3. The significance of this study is not clearly provided. 

We have better highlighted the significance of the study at the beginning of the
abstract, as follows:

“Documenting the perceptions and experiences of frontline healthcare workers during a
sanitary crisis is key to reinforce healthcare systems. We identify the determinants
of quality of working life (QWL) among night-shift healthcare workers (NSHW) in
public hospitals in Paris shortly after the first-wave of the COVID-19
pandemic.”

4. No information is given on the first wave of pandemic? Why the assessment was done
after first wave? What is the significance of having this study after first wave? 

In France, the first wave of the COVID-19 pandemic lasted from March to May 2020. The
ALADDIN survey was performed shortly after from 15 June to 15 September 2020), which
enabled us to reach NSHW after the initial urgent period, once they were less in the
heat of the action, more available to answer the survey, and to have a little
hindsight to report their perceptions. 

We have modified the following sentence of the Methods section:

“One of the main objectives was to document NSHW’s QWL (i.e. perceived quality of
life at work) and its correlates shortly after the first wave of the COVID-19
pandemic (March to May 2020), once healthcare workers were more available to
participate in the survey.”

5. “The ALADDIN cross-sectional survey (15 June 2020 to 15 September 2020) 76 was
conducted among NSHW in the 39 hospitals of the Assistance Publique - Hôpitaux de
Paris (AP-HP).” How the hospitals are selected? And any inclusion/ exclusion
criterias for the selection? 

The survey targeted NSHW in all public hospitals of the Parisian area. It was
conducted among all 39 hospitals of the Assistance Publique - Hôpitaux de Paris
(AP-HP), with no specific inclusion/exclusion criteria at the level of
hospitals.

We have specified this in the Methods section of the revised manuscript, as
follows:

“The ALADDIN cross-sectional survey (15 June 2020 to 15 September 2020) was conducted
among NSHW in public hospitals in Paris. It included all 39 hospitals of the
Assistance Publique - Hôpitaux de Paris (AP-HP).” 

6. What is the response rate? This may lead to information bias when the response
rate is too low. 

The web survey was available to all NSHW of the AP-HP hospitals. The response rate
was approximately 11.5% (1387 /12,000), which was close to our initial objective
(1200/12,000). Independently of the response rate, it has to be noted that data
weighting and calibration enabled us to work on a representative dataset in terms of
age, sex, and professional category. 

7. Few of the variables, i.e., “physical activity”, “Change in weight since works at
night”, “Travel time to work” are unclear to me. What are the operational
definitions for these variables. Table 3: 0.7 hours in the travel time means? 

Information on physical activity was collected using the following item:

“Do you practice physical activity?”

Similarly, healthcare workers were asked if they perceived their weight to not have
changed, increased or decreased since the beginning of night-shift work.

“Travel time to work” relates to the duration of the home-work one-way commute.

We have modified the label of these variables in the tables, as follows:

- “Physical activity” has been changed in “Practice of any physical activity”.

- “Change in weight since works at night” has been changed to “Perception of a change
in weight since working at night”.

- “Travel time to work” has been changed to “Travel time to work (home-work one-way
commute)”.

0.7 hours of commute means that the NSHW spends nearly three quarters of an hour for
a one-way commute. We have converted travel times to minutes in the table.

8. Page 29, Line 238: “social and professional recognition ” is unclear to me. Please
explain. 

We have added a description of variables related to social and professional
recognition in the Methods section of the revised manuscript, as follows (Data
collection paragraph):

“NSHW’s perceptions regarding their social and professional recognition were assessed
using items related to under-estimation of night-shift work by colleagues, loved
ones, and patients; perceptions of the importance of night missions and of workload
during the night; and feeling valued by the general population as a NSHW during the
pandemic.”

We have also enriched the discussion of workers’ recognition during the pandemic, as
follows:

“Professional recognition also includes feeling supported by peers. In the ALADDIN
survey, 64.7% of NSHW reported that night-shift work is often or always
under-estimated by colleagues working during the day, and this perceived stigma had
a significant detrimental effect on QWL. These findings highlight the need to
develop interventions to improve communication, sharing of experiences, and support
between day-shift and night-shift hospital healthcare workers. Such interventions
can reinforce the sense of community among healthcare workers, and have the
potential to improve NSHW’s daily experience in the workplace.”

9. Table 1 is poorly presented. 

We have made a short description of each group of variables presented in Table 1. A
more detailed description would lead to redundancies between text and tables.

Of note, as recommended by Reviewer 1, we have modified Table 1 to improve
readability: it has been split into six different sub-tables, each representing a
group of variables. This allows a correspondence with paragraphs in the Results
section. 

10. Page 30, Line 283: “The lack of reference values for the WRQoL scale in France
also limits the discussion of results.”. What is the importance of having reference
values in this study? 

We refer here to the fact that “WRQoL” norms were developed using data collected in
UK National Health Services, and not in France. As this point was already raised
earlier in the Discussion, we have removed the sentence page 3 line 283.

11. Page 30, Line 263: “These differences were however of modest magnitude and did
not exceed 3 points in QWL scores.” Is 3 points as a cutoff value to determine the
difference between groups? Please provide references. 

To our knowledge, the minimum important difference in WRQoL scores is not documented
among healthcare workers. We have added the following sentence in the
Discussion:

“Further research is needed to determine if such a magnitude exceeds the minimum
important difference for the WRQoL scale.”

12. Page 31, Line 290: “Findings from the AP-HP ALADDIN survey contribute to increase
the body of knowledge about these key issues, which are central to set up efficient
strategies to reinforce healthcare systems.” What are the examples of strategies
that could be recommended by the authors based on the findings of this study. 

We have added examples of strategies as follows:

“Such strategies should include interventions aiming to improve recognition, reduce
stigma related to night-shift work, and to improve information and communication
between the different groups of healthcare workers.”

13. The findings of Table 2 is poorly discussed.

Findings in Table 2 (distribution of QWL scores) are difficult to discuss due to a
lack of data in the literature. We have raised this point as follows:

“There is a lack of published studies on QWL conducted among healthcare workers,
especially in France. We identified only one recent survey, which was also based on
the WRQoL scale (40). QWL level observed in ALADDIN was lower than that found in
this recent survey, conducted among 2,040 French anesthesiologists (median [IQR]
WRQoL full-scale score: 71 [63-78] versus 77 [66–85]) (40). This difference is
likely to be related to the study period, as the latter survey was performed before
the beginning of the COVID-19 pandemic (January to June 2019). It may also be
related to the diversity of professional categories participating in ALADDIN,
presenting different levels of QWL.”

Other results presented in Table 2 (differences in scores associated with each
dimension of QWL) are discussed as follows: 

“Executives showed both the best overall QWL and higher scores in the “Home-work
interface” and “Job and career satisfaction” dimensions of QWL, compared with that
of other professional categories. Along with older age, correlated with less
domestic responsibilities related to childcare, a longer experience of night-shift
work may explain the greater ability of executives to find the right balance between
their professional and personal lives. By contrast, executives (together with
midwives) presented a low QWL score in the “Stress at work” dimension, revealing
higher levels of stress than other professional categories. Interestingly, midwives
reported the lowest QWL related to working conditions. Further research should thus
be performed to identify midwives’ specific needs and expectations to both improve
their QWL and prevent psychosocial risks (42).”

to reviewers 20janv2022.docx
---

## [Decision Letter · Decision Letter 1]

8 Mar 2022

Social and professional recognition are key determinants of quality of life at work
among night-shift healthcare workers in Paris public hospitals (AP-HP ALADDIN
COVID-19 survey)

PONE-D-21-27355R1

Dear Dr. Marcellin,

We’re pleased to inform you that your manuscript has been judged scientifically
suitable for publication and will be formally accepted for publication once it meets
all outstanding technical requirements.

Kind regards,

Florian Fischer

Academic Editor

PLOS ONE

Additional Editor Comments (optional):

Reviewers' comments:

Reviewer's Responses to Questions

**Comments to the Author**

1. If the authors have adequately addressed your comments raised in a previous round
of review and you feel that this manuscript is now acceptable for publication, you
may indicate that here to bypass the “Comments to the Author” section, enter your
conflict of interest statement in the “Confidential to Editor” section, and submit
your "Accept" recommendation.

Reviewer #2: All comments have been addressed

2. Is the manuscript technically sound, and do the data
support the conclusions?

Reviewer #2: Yes

3. Has the statistical analysis been performed
appropriately and rigorously? 

Reviewer #2: Yes

4. Have the authors made all data underlying the
findings in their manuscript fully available?

Reviewer #2: Yes

5. Is the manuscript presented in an intelligible
fashion and written in standard English?

Reviewer #2: Yes

6. Review Comments to the Author

Reviewer #2: (No Response)

7. PLOS authors have the option to publish the peer
review history of their article (what does this mean?). If published, this will
include your full peer review and any attached files.

If you choose “no”, your identity will remain anonymous but your review may still be
made public.

**Do you want your identity to be public for this peer review?** For
information about this choice, including consent withdrawal, please see our
Privacy Policy.

Reviewer #2: **Yes: **YIN CHENG LIM

---

## [Editor Report · Acceptance letter]

30 Mar 2022

PONE-D-21-27355R1 

Social and professional recognition are key determinants of quality of life at work
among night-shift healthcare workers in Paris public hospitals (AP-HP ALADDIN
COVID-19 survey) 

Dear Dr. Marcellin:

I'm pleased to inform you that your manuscript has been deemed suitable for
publication in PLOS ONE. Congratulations! Your manuscript is now with our production
department. 

Kind regards, 

on behalf of

Dr. Florian Fischer 

Academic Editor

PLOS ONE